# Assessing the Prognosis of Patients with Myelodysplastic Syndromes (MDS)

**DOI:** 10.3390/cancers14081941

**Published:** 2022-04-12

**Authors:** Annika Kasprzak, Kathrin Nachtkamp, Norbert Gattermann, Ulrich Germing

**Affiliations:** Department of Hematology, Oncology and Clinical Immunology, Heinrich-Heine University, 40225 Duesseldorf, Germany; kathrin.nachtkamp@med.uni-duesseldorf.de (K.N.); gattermann@med.uni-duesselddorf.de (N.G.); germing@med.uni-duesseldorf.de (U.G.)

**Keywords:** myelodysplastic syndromes, prognosis, chronic myelomonocytic leukemia

## Abstract

**Simple Summary:**

Myelodysplastic syndromes are a group of clonal disorders originating from hematopoietic stem and progenitor cells. Due to their heterogenous nature, prognostic stratification as well as therapeutic management remain a challenge. For the majority of MDS patients, treatment strategies are risk-adapted. Current prognostic scoring systems rely on a number of key factors which help to assess the individual prognosis. Despite a number of recent advances, the integration of important patient- and disease-related parameters still falls short. This article highlights the most important scoring systems, summarizes their potential use in clinical practice, and addresses important questions on the assessment of prognosis in patients with myelodysplastic syndromes and chronic myelomonocytic leukemia.

**Abstract:**

Prognostic stratification in patients with myelodysplastic syndrome (MDS) relies on a number of key factors. Combining such patient-related and disease-related prognostic parameters into useful assessment tools remains a challenge. The most widely used scoring systems include the international prognostic scoring system (IPSS), the revised IPSS (IPSS-R), the World Health Organization (WHO) Prognostic Scoring System (WPSS), and the new molecular IPSS (IPSS-M). Similar to the IPSS-R and the IPSS-M, the chronic myelomonocytic leukemia (CMML) prognostic scoring system (CPSS) and the CPSS molecular (CPSS-mol) are powerful and reliable prognostic tools that help to assess the individual prognosis of patients with CMML. The well-established prognostic assessment of MDS and CMML may be further augmented by additional disease-related parameters, such as somatic mutations, or patient-related factors, such as comorbidities. In this article, we briefly describe useful prognostic scoring systems for myelodysplastic syndromes and identify some open questions that require further investigation.

## 1. Introduction

Correct diagnosis and reliable prognostic assessment are critical for individualized clinical decision-making in patients with myelodysplastic syndromes (MDS). Treatment aims to improve the patients’ well-being and life expectancy. The clinical course of MDS may vary significantly. Patients with low-risk disease have almost the same life expectancy as the age- and gender-adapted non-MDS population. On the other hand, high-risk MDS, with its propensity to leukemic transformation and therapy resistance, may shorten survival time to less than a year. In order to adapt the intensity of treatment to the severity of the disease, a proper risk assessment is needed. Therefore, prognostic parameters and/or prognostic scoring systems have been developed to separate groups of patients that differ in terms of the median survival time. Here, we discuss established and evolving tools for prognostic assessment of patients with MDS.

To the best of our knowledge, most of the literature on risk assessment in MDS is based on clinical data reflecting the course of disease in patients who did not receive potentially disease-modifying treatments, such as induction chemotherapy, hypomethylating agents (HMA), or allogeneic stem cell transplantation (alloSCT). In particular, the IPSS was designed based on a patient cohort that only received the best supportive care. As the proportion of patients receiving such treatment has increased over the years, it may be argued that older prognostic scoring systems are no longer suitable for the current situation. However, it is also arguable whether any treatment, apart from alloSCT, has a major influence on MDS prognosis [1,2]. According to the Düsseldorf MDS Registry, patients’ prognoses have not changed substantially over the last two decades, regardless of the MDS type and treatment [3].

Therapeutic options specifically targeting disease biology are rare [4]. The only examples are lenalidomide for low-risk MDS with *del(5q)*, erythropoiesis-stimulating agents (ESA) for patients with only modestly elevated levels of endogenous erythropoietin, and luspatercept for lower-risk MDS with ring sideroblasts and/or *SF3B1* mutation.

MDS prognosis is primarily influenced by disease-related parameters, such as low blood cell counts. In particular, neutropenia and thrombocytopenia are the predominant causes of death in MDS. In order to describe the prognosis of MDS patients, several endpoints are used. The most important parameter is overall survival (OS). In addition, the time to the development of AML, the cumulative risk of AML, or leukemia-free survival (LFS) is usually estimated. In the MDS patient population as a whole, leukemic transformation certainly shortens OS. However, in patients with advanced MDS, as indicated by an elevated blast count, transformation into AML does not significantly impact OS because the major causes of death are the same in advanced MDS and AML, namely, infection and bleeding as a result of hematopoietic insufficiency. In patients with lower-risk MDS, though, the development of AML is a decisive event in terms of prognosis [5].

## 2. Prognostic Parameters in MDS

Prognostic parameters can be divided into disease-related and patient-related factors. Disease-related factors, such as peripheral blood counts or cytogenetics, reflect the biology of the underlying bone marrow disease, whereas patient-related factors, such as performance status and comorbidities, are at least partly independent of MDS biology. However, both categories can overlap and influence each other. For instance, while MDS-related anemia may be tolerable for a patient without cardiovascular and/or pulmonary disease, similar hemoglobin levels may cause symptoms in a patient with cardiac comorbidity and thus become a pivotal prognostic factor.

Disease-related prognostic parameters such as anemia and thrombocytopenia [6], as well as neutropenia and lymphocytopenia [7], reflect hematopoietic insufficiency and directly contribute to the major causes of death, namely, infections and bleeding [8]. However, increased rather than decreased cell counts may also have a prognostic impact: Monocytosis in the peripheral blood of patients with classical MDS may be associated with worse survival rates [9].

Parameters reflecting increased cell turnover, such as elevated LDH, thymidine kinase, and beta2-microglobulin, have long been known as prognostic parameters [6,10]. Serum ferritin (SF), reflecting storage iron, has also been shown to be a prognostic factor. Elevated SF indicates transfusional iron overload, which is prognostically relevant because it is a surrogate marker for hematopoietic insufficiency as well as iron-related organ damage. If detected at diagnosis, elevated SF may result from increased intestinal iron uptake due to down-regulation of hepcidine as a consequence of ineffective erythropoiesis [11,12,13,14]. It is plausible that parameters indicating inflammation and/or an impaired bone marrow microenvironment, such as increased C-reactive protein (CRP), erythrocyte sedimentation rate (ESR), or S100-A9 [15], have a prognostic impact as well. In addition, the assessment of marrow cellularity by cytology, or more reliably by histopathology, yields prognostic information: patients with a hypercellular marrow, partly arising from an unsuccessful attempt at compensating for peripheral cytopenias, fare worse than patients with a normo- or hypocellular marrow [16,17,18]. Marrow fibrosis has prognostic implications since it is associated with more pronounced hematopoietic insufficiency, a higher risk of clonal evolution, and more frequent progression to AML [19]. The degree of dysplasia in blood and bone marrow can also be utilized for prognostication [20]. Multilineage dysplasia is related to impaired differentiation, intramedullary apoptosis, and hematopoietic insufficiency.

The percentage of bone marrow blasts is one of the most important prognostic parameters in MDS and is related to the degree of hematopoietic insufficiency, presence of clonal evolution, loss of apoptotic activity, and risk of progression to AML [6,20,21,22]. As the percentage of marrow blasts might reflect the size of the (sub)clone that is most severely compromised in terms of cell differentiation, it is appropriately used for staging and prognostication. However, a small subclone with a low variant allele frequency (VAF) might exist, which can have a higher potential for leukemic transformation. Chromosomal and molecular genetic aberrations also have a substantial impact on survival and the risk of AML evolution. Regarding somatic mutations, not only their presence but also their variant allele frequency (VAF) can yield prognostic information. Aberrant expression of genes such as WT1 [23,24,25,26] and aberrant findings on proteomic analysis are being explored as prognostic markers [27,28]. The WHO classification, which is currently being revised, takes into account the medullary blast count, multi- vs. single-lineage dysplasia, cytogenetic information (*del5q*), and a single molecular genetic marker (*SF3B1* gene mutation).

All disease-related parameters are dynamic and can deteriorate during the course of the disease, thus indicating a change in prognosis due to aggravated bone marrow failure, deteriorating thrombocytopenia [29,30], greater transfusion requirement [31], clonal evolution [23,32], and increased risk of leukemic transformation. The dynamic nature of prognostic factors implies that it may be necessary to repeatedly reassess the patient’s prognosis [33].

Regarding patient-related prognostic parameters, age at the time of diagnosis is certainly important. However, the survival impact of age decreases with the severity of MDS [34]. Among patients with low-risk MDS, according to IPSS-R, age is a strong prognostic factor, whereas, among patients with high-risk MDS, prognosis only marginally differs between older and younger individuals. This also applies to comorbidities [35,36]. However, serious comorbidities, such as severe heart failure or metastatic cancer, can limit the patient’s prognosis, irrespective of the prognostic category of MDS.

Other factors that may influence the prognosis of MDS are a history of preceding conditions, such as long-term immunosuppression; an underlying inherited hematological disease (Fanconi anemia, etc.); and, most importantly, a history of mutagenic treatment, i.e., radiotherapy and/or chemotherapy) [37]. Therapy-related myelodysplastic syndromes (t-MDS) are defined as MDS occurring after the application of cytotoxic chemotherapy and/or radiation in the context of a malignant or even non-malignant disease. They belong to the group of therapy-related myeloid neoplasms (t-MNs). Most prognostic scoring systems which have been developed, exclude patients with a therapy-associated myeloid neoplasm. Kündgen et al. [37] investigated the prognostic effect of the most common scores in a large cohort of t-MDS patients. All the investigated scoring systems, including the IPSS, IPSS-R, and WPSS, were able to correctly discriminate different risk groups. However, the performance and prognostic power of those scores were inferior compared to prognostication in patients with primary MDS. The IPSS-R and the WPSS were the most exact scores, subdividing patients with t-MDS into groups with varying prognoses. Patients with therapy-associated MDS have inferior OS compared to de novo MDS, partly due to adverse chromosomal and molecular genetic features and partly due to the patients’ general medical condition being compromised by a prior malignancy and its treatment. Amongst molecular genetic aberrations, mutations in *TP53* proved to be of utmost importance, as about 50% of t-MDS patients have aberrations in this gene. MDS patients presenting with TP53 mutations are classified as high-risk patients [38]. Furthermore, patients with t-MDS present with a complex karyotype due to the exposition of mutagenic substances, equally worsening the prognosis. Prognostic scoring systems, including cytogenetic and molecular genetic aberrations, are presented below.

## 3. Prognostic Scoring Systems

In order to integrate different prognostic parameters into scoring systems, multivariate analyses of sizeable data sets were performed (Table 1). This resulted in numerous prognostic scoring systems [5], including blood cell counts, blast percentages, LDH, signs of dysplasia, karyotypes, molecular genetic findings, and age and gender (Figure 1). It became clear that certain parameters, namely, cell counts, transfusion need, medullary blast count, and cytogenetic findings, contribute prognostic information independently and can therefore serve as useful components of prognostic tools. The international prognostic scoring system (IPSS) [39] and its successor, the revised version of the IPSS (IPSS-R) [40], are robust prognostic scoring systems that have been validated repeatedly [41]. Both include the degree of cytopenias, bone marrow blast percentage, and cytogenetic risk categories. The WHO-adapted prognostic scoring system (WPSS) replaced hemoglobin values with transfusion requirements [42]. The well-established association between transfusion need and prognosis has multiple explanations, including anemia-related cardiac problems, transfusional iron overload, and transfusion requirement being a surrogate marker for bone marrow failure. The WPSS was the first score that was applicable at various time points during the course of the disease. This is mainly attributable to the emphasis on karyotype and the need for transfusion. Chromosomal findings, which rank high in the WPSS, reflect clonal evolution in 15–20% of MDS patients. In addition, the onset of transfusion need, which occurs in the majority of MDS patients, reflects the prognostically relevant deterioration of bone marrow function [23,32]. Furthermore, the WPSS is sensitive to changes in the WHO subtype of the disease, caused by an increasing medullary blast count or aggravation of dysplasia. The issue of time-dependant changes in prognostic parameters and the need for dynamic scoring systems were analyzed and explained by Pfeilstöcker et al. [33].

The application of the IPSS and the IPSS-R remains limited to the initial MDS diagnosis. The WPSS, however, provides the advantage of a dynamic assessment during the course of the disease and takes into consideration time-dependent changes of the most important prognostic variables, such as the WHO diagnostic classification, karyotype, and transfusion requirement [43].

There are several parameters that are not included in the above-mentioned scoring systems but can still be helpful in individual clinical cases. The non-consideration of parameters such as LDH and bone marrow fibrosis is mainly due to the fact that these parameters were not broadly assessed at initial diagnosis, often because they are not part of the routine diagnostic workup in some countries.

Scoring systems have also been specifically developed for patients with MDS/MPN overlap syndromes, such as chronic myelomonocytic leukemia (CMML). Such scores are methodically very similar to MDS scores but include CMML-specific parameters. In the CMML prognostic scoring system (CPSS) [44], cytogenetic risk categories differ from the IPSS-R; the proliferative variant of CMML is addressed by considering leukocyte counts >13,000/µL and medullary blast count, and hematopoietic insufficiency is reflected by transfusion requirements. In this way, an equally robust and validated tool has emerged that is capable of separating five CMML risk groups throughout the course of the disease. With growing knowledge of the prognostic impact of somatic mutations and their characteristic profile in CMML, a refined CPSS-mol [45] was developed. As mutations in *ASXL1*, *NRAS*, *RUNX1*, and *SETBP1* were proven to have an independent prognostic impact on OS and risk of progression to AML, each mutation was weighted differently and integrated separately into the CPSS-mol. The CPSS-mol identifies very-low-risk CMML patients who virtually never develop AML. It also identifies patients of the former CPSS low- and intermediate-risk groups who actually have inferior survival compared to patients with high- and very-high-risk CMML.

Similarly, the international working group for the prognosis of MDS (IWG-PM) recently integrated molecular data into the IPSS-R, resulting in the IPSS m [26]. The cytogenetic risk groups of the IPSS-R, based on the seminal work of Schanz et al. [46], were left unchanged because no better classification could be derived from the available data sets. Hemoglobin values, platelet counts, and medullary blast percentages were included as continuous variables, and the absolute neutrophil count was eliminated because it was prognostically less robust (Table 2).

Evidence of somatic mutations in *TP53*, *MLL*, *FLT3*, *SF3B1*, *del*(*5q*), *NPM1*, *RUNX1*, *NRAS*, *ETV6*, *IDH2*, *CBL*, *EZH2*, *U2AF1*, *SRSF2*, *DNMT3A*, *ASXL1*, *KRAS*, and *SFRB1* was integrated into the IPSS-M, with individual weight attributed to each variable. Moreover, the number of mutated genes (0 vs. 1 vs. ≥2) from a list of 15 further genes (*BCOR*, *BCORL1*, *CEBPA*, *ETNK1*, *GATA2*, *GNB1*, *IDH1*, *NF1*, *PHF6*, *PPM1D*, *PRPF8*, *PTPN11*, *SETBP1*, *STAG2*, and *WT1*) was integrated as an additional variable. The aforementioned genes vary in prognostic power and may be subdivided into good- and poor-risk genes. The by far most important genetic aberration is a mutation in *TP53.* Patients with MDS who have biallelic *TP53*-mutations have worse outcomes, such as therapy-refractory disease, rapid progression into AML, and shorter OS, compared to patients with monoallelic mutations [25]. Bernard et al. proved that patients with biallelic vs. monoallelic *TP53* significantly differ in OS: patients in the multi-hit state of *TP53* have a median OS of only 8.7 months, while patients with monoallelic state were found to have a median OS of 2.5 years. Furthermore, patients in the multi-hit state suffer from transformation into AML more often. As this work shows, the correct diagnostic and prognostic stratification of MDS patients requires assessing the *TP53* state; the *TP53* allelic state received special consideration within the new IPSS-M to identify true high-risk MDS patients. The model was furthermore adjusted by three relevant variables influencing the prognosis, namely, gender (males have an inferior prognosis), age as a continuous variable, and MDS subtype (primary vs. therapy-associated). Wisely, these three parameters were not integrated into the mathematical model, resulting in a model that relies solely on disease-associated parameters. The new IPSS-M defines six risk groups that differ significantly in terms of OS and LFS over the entire course of the disease. These risk groups are more homogeneous than those in the IPSS-R. A web-based application was designed to facilitate risk calculation.

Table 3 lists the major prognostic scoring systems for MDS or CMML and the prognostic parameters utilized. The initial IPSS emphasized the medullary blast count and three cytogenetic risk groups but neglected blood cell counts. The number of cell lineages affected by cytopenia was considered, but not the respective degree of cytopenia. The WPSS focused on chromosomal findings, integrated the dimension of dysplasia, and considered the need for treatment. The IPSS-R implemented five thoroughly evaluated cytogenetic risk groups, refined the medullary blast categories, including a low (practically normal) blast count category of 0–2%, and took into account the degree of hematopoietic insufficiency reflected by peripheral blood cell counts. Finally, the very sophisticated IPSS-M was developed, which can integrate the results of extended mutation analysis. However, we should like to point out that molecular analysis is not a prerequisite for the prognostic assessment of MDS patients. If molecular testing is not available, the IPSS-R and WPSS for patients with MDS, and the CPSS for patients with CMML, provide very useful prognostication with regard to OS and LFS.

## 4. Scores Addressing Patient-Related Prognostic Parameters

In addition to disease-related factors, comorbidities are important for the prognostic assessment of MDS patients. Relevant comorbidities usually lead to impaired organ function, diminished quality of life, and the need for therapeutic intervention. Comorbidities may not only limit the patient’s prognosis through comorbidity-related complications but also by precluding intensive treatment of MDS. Cardiac and renal failure, for example, prevent the patient from undergoing allogeneic stem cell transplantation. The most frequent problems are cardiac, hepatic, and renal comorbidities, as well as second primary tumors [35]. Numerous other comorbidities are too rare to be included in scoring systems. The MDS working groups in Pavia and Düsseldorf closely collaborated to study the prognostic impact that comorbidities have on the course of MDS and developed a useful tool for risk assessment. The MDS comorbidity index (MDS-CI) includes five major categories: cardiac diseases (2 points); diseases of liver, kidney, and lung; as well as malignancies (each 1 point). The low-risk group (0 points), intermediate-risk group (1–2 points), and high-risk group (>2 points) have different median survival times, independent of the patients’ IPSS score [47].

Other comorbidity scores may be applied to patients with MDS, such as the hematopoietic stem cell transplantation comorbidity index (HCT CI) [48]. This score is meant to be used prior to alloSCT to determine whether a patient is suitable for the procedure. Uncritical use of this score may lead physicians to withhold intensive treatment from a patient who might need it for high-risk MDS.

Finally, some scores, such as the Texas Score, combine disease-related and patient-related parameters [49]. This approach can be useful in clinical practice, but it hampers the ability to attribute clinical outcomes to biological disease characteristics versus age and comorbidities.

## 5. Conclusions and Future Directions

The identification and evaluation of prognostic parameters are ongoing. Consideration of low lymphocyte counts [7], monocytosis [7], and dynamic evolution of cytopenia [29,30], as well as flow cytometric investigations [50] analysis or clonal evolution [32], and measurement of WT1 expression [24], may be used in addition to the established scoring systems in the future.

Some open questions regarding prognostication in MDS require further investigation.

(1)Prognostic parameters may differ between treated and non-treated patients.(2)Prognostic factors relevant for the natural course of disease must be differentiated from predictors of response to treatment.(3)If successful treatment options are available, predictors of response may also serve as prognostic factors for survival.(4)Specific prognostic parameters may exist for patients with therapy-related MDS [37].

Prognostic scoring systems in MDS are usually applicable to treated patients as well as untreated patients since most therapeutic interventions do not result in extended survival. Furthermore, unfavorable prognostic parameters often retain their negative impact in patients receiving potentially disease-modifying treatment, i.e., hypomethylating agents or alloSCT.

There are not many predictors of treatment success in MDS. Examples are the achievement of complete remission, or at least considerable downsizing of the del(5q) clone, in patients treated with lenalidomide [51], evidence of SF3B1 mutation in patients going to receive Luspatercept [52], and relatively low endogenous erythropoietin (EPO) [53] levels in patients going to receive an ESA or an analog, respectively. Reliable predictors of response to HMAs are largely lacking [54]. Besides identifying prognostic and predictive parameters, another important goal is to identify and utilize markers of minimal residual disease, in particular after allogeneic stem cell transplantation [54].

## Figures and Tables

**Figure 1 cancers-14-01941-f001:**
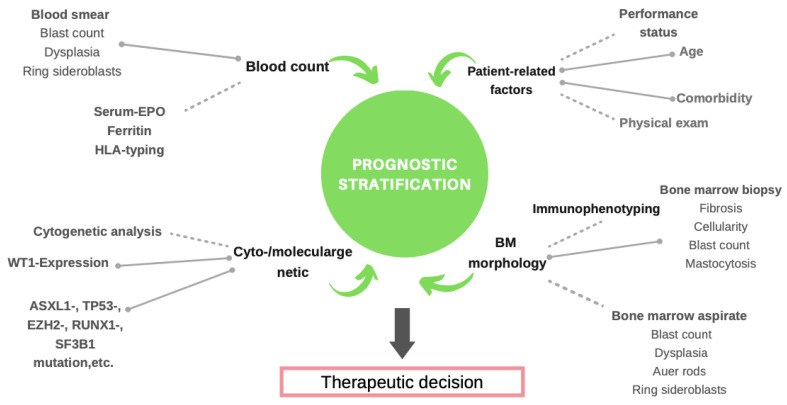
Clinical, morphological, cytogenetical, and molecular features in prognostic stratification of MDS patients.

**Table 1 cancers-14-01941-t001:** Key prognostic parameters in patients with MDS.

Parameter	Disease-Related	Patient-Related	Used in Major Scoring Systems	Comments
Age		x		Impact is limited in HR-MDS
Gender		x		Impact is limited in HR- MDS
Comorbidities		x		Partly interacts with disease characteristics
Therapy-related MDS (t-MDS)	x	x		Related to HR-genetic findings and previous malignancies and their mutagenic treatment
Transfusion need	x	x	x	Reflects hematopoietic insufficiency as well as comorbidities
Low cell counts (Hb, PLT, ANC, lymphocytes)	x		x	Indicates hematopoietic insufficiency. Related to causes of death
High WBC	x		x	Indicates clonal proliferation
Declining cell counts	x			Indicates disease progression
pB blasts	x			Indicates disease progression, including leukemic transformation
Bm blast percentage	x		x	Indicates disease progression, including leukemic transformation
LDH, thymidine kinase, β2-MG	x			Confirmed impact on prognosis, independent of genetic findings
ESR, CRP, S100	x			Indicates inflammatory activity in the bone marrow
Serum ferritin, hepcidine	x			Reflects transfusional iron overload and duration of erythropoietic insufficiency
Flow cytometry anomalies	x			Indicates dysplasia and size of the malignant clone
Marrow cellularity	x			Reflects proliferation in the hematopoietic system
Marrow fibrosis	x			Indicates progression and marrow failure
Multilineage dysplasia	x			Indicates the cell lineages involved
Cytogenetics	x		x	Robust parameters may influence treatment decisions (del5q) and mutation of SF3B1
Somatic mutations	x		x	Reflects driver mutations and degree of clonal instability
Clonal evolution	x			Indicates disease progression

**Table 2 cancers-14-01941-t002:** The IPSS-M: major prognostic categories and its parameters.

Category	Parameters	Additional Information
**CLINICAL PARAMETERS**	Marrow blasts	Continous clinical parameters
	Platelets	
	Hemoglobin	
**IPSS-R CYTOGENETIC RISK CATEGORIES**	Very low	As applied within the IPSS-R
	Low	
	Intermediate	
	High	
	Very high	
**GENETIC MUTATIONS**	16 predictive gene mutations	Individual weights attributed to each variable
	15 additional genes	One feature representing the number of mutations from this group

**Table 3 cancers-14-01941-t003:** Major prognostic scoring systems for patients with MDS or CMML (dark green to light green: most important to least important key factors in each soring tool).

Score	BM Blasts	WHO Class	Cyto Genetics	Hb	Plt	ANC	WBC	Transf. Need	Molec. Findings
**IPSS 1997**	yes 4 cat.	no	yes 3 cat.	yes 2 cat.	yes 2 cat.	yes 2 cat.	no	no	no
**WPSS 2007**	indirect	yes 4 cat.	yes 3 cat.	no	no	no	no	yes 2 cat.	no
**IPSS-R 2012**	yes 4 cat.	no	yes 5 cat.	yes 3 cat.	yes 3 cat.	yes 2 cat.	no	no	no
**IPSS-M 2022**	yes cont.	no	yes 5 cat.	yes cont.	yes cont.	no	no	no	yes many cat.
**CPSS 2013**	yes 2 cat.	no	yes 3 cat.	no	no	no	yes 2 cat.	yes 2 cat.	no
**CPSSmol 2016**	yes 2 cat.	no	yes 3 cat.	no	no	no	yes 2 cat.	yes 2 cat.	yes 4 cat.

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
