# Peer review of "Assessing the Prognosis of Patients with Myelodysplastic Syndromes (MDS)"

_cancers, 2022, doi:10.3390/cancers14081941_

Round 1

Reviewer 1 Report

This is a comprehensive review addressing all the evolution of major prognostic indicators in MDS and CMML. 

Specific comments:

  1. The authors have did not give adequate details on the molecular prognostic models of MDS. Tabulating the IPSS-M will improve clarity.
  2. The authors should describe the recently published work on the molecular prognostication of MDS and CMML. They include: Naza et al. JCO 2021; Bersanelli et al. JCO 2021 39:1223-1233. 
  3. Prognostic indications in therapy-related MDS should be specifically discussed. 
  4. A summary with with algorithmic approach to prognostication in MDS and CMML involving clinical, pathologic, CG and molecular features, should be included using a flowchart/table.

Author Response

Dear Reviewers, dear Editor,

thank you for taking the time to assess our manuscript. The constructive advice has substantially improved our paper. Please find attached our responses to the comments.

Reviewer 1: The authors have did not give adequate details on the molecular prognostic models of MDS. Tabulating the IPSS-M will improve clarity.
Authors: We thank the reviewer for pointing this out. To clarify the IPSS-M we added a table with all prognostic parameters.

Reviewer 1: The authors should describe the recently published work on the molecular prognostication of MDS and CMML. They include: Naza et al. JCO 2021; Bersanelli et al. JCO 2021 39:1223-1233. 
Authors: We thank the reviewer for this suggestion. We added the important work into our manuscript.

Reviewer 1: Prognostic indications in therapy-related MDS should be specifically discussed. 

Authors: We appreciate the suggestion and added a section on t-MDS and the prognostic impacts of various parameters in this context.

Reviewer 1: A summary with with algorithmic approach to prognostication in MDS and CMML involving clinical, pathologic, CG and molecular features, should be included using a flowchart/table

Authors: We added a figure which illustrates the most important variables, which have to be taken into account for risk stratification in MDS patients.

Reviewer 2: Line 55-56. “To our knowledge, most of the literature on risk assessment in MDS is based on clinical data reflecting the course of disease in patients who did not receive potentially disease-modifying treatments”. I would revise this statement to mention that IPSS was developed in an era of supportive care. Most of the recent prognostic systems including R-IPSS and IPSS-M have been tested in patients who received standard therapies.
Authors: Thank you for pointing this out. We changed this sentence.

Reviewer 2: Anemia does not promote development of heart failure (line 88), but it can cause symptoms in a patient without good cardiopulmonary reserve.
Authors: This is correct. We changed our choice of words.

Reviewer 2: Line 94: MDS pts with monocytosis may have CMML. I advise caution in including such a statement.
Authors: We have softened our message.

Reviewer 2: Line 97-98: The statement on the low blast percentage (1-5%) and AML risk is not well-supported so should be removed.
Authors: We agree that this statement is not entirely validated. We removed it.

Reviewer 2: All the clinicopathologic variables discussed between lines 99 and 117 have limited prognostic value when factored against the R-IPSS parameters and the molecular mutations. Their limitation has to be stated.
Authors: We have modified this paragraph to outline the limitations.

Reviewer 2: Lines 121-122 “As the percentage of marrow blasts roughly reflects the size of the (sub)clone that is most severely compromised in terms of cell differentiation, it is appropriately used for staging and prognostication.” This statement is not accurate biologically and clinically. There may be a very small subclone with low VAF, which can have a higher potential for leukemic transformation.
Authors: We thank the reviewer for pointing this out. We changed our choice of words in this paragraph.

Reviewer 2: Lines 130-132. “As it incorporates the disease-related factors with the highest prognostic impact, the WHO classification can be used as an approximative prognostic tool”. This sentence does not add anything.
Authors: This sentence was removed.

Reviewer 2: Lines 133-138: How would you assess prognosis dynamically when IPSS has never been validated prospectively through the disease course of a patient?
Authors: It is correct, that the IPSS is not a dynamic score. However, the WPSS offers the possibility of a dynamic assessment. We added further explanations to clarify this.

Reviewer 2: Lines 139-148: Age and gender effects are very controversial. I recommend removing these statements.
Authors: We agree with the reviewer and removed these statements.

Reviewer 2: Line 150-156. The authors need to explain therapy-related MDS further including its adverse risk cytogenetic/molecular characteristics (references: 34298596, 30670442).
Authors: We elaborated the topic of t-MDS and added the suggested references.

Reviewer 2: Lines 210-223: Authors should discuss good- and poor-risk genes of IPSS-M with a special focus on TP53.
Authors: To further clarify the IPSS-M we discussed molecular aberrations, especially TP53 and added a table with the prognostic variables of the IPSS-M.

Reviewer 2: Line 291. TPO level is not something we can measure in a routine clinical lab and it is also not validated as a predictive tool.
Authors: Our lab is able to measure TPO levels, however this is not true for every hospital. We have therefore clarified potential limitations in clinical routine.

Reviewer 2 Report

  1. Line 55-56. “To our knowledge, most of the literature on risk assessment in MDS is based on clinical data reflecting the course of disease in patients who did not receive potentially disease-modifying treatments”. I would revise this statement to mention that IPSS was developed in an era of supportive care. Most of the recent prognostic systems including R-IPSS and IPSS-M have been tested in patients who received standard therapies.
  2. Anemia does not promote development of heart failure (line 88), but it can cause symptoms in a patient without good cardiopulmonary reserve.
  3. Line 96: Kasprzak`s paper should be listed as a reference rather than writing it openly.
  4. Line 94: MDS pts with monocytosis may have CMML. I advise caution in including such a statement.
  5. Line 97-98: The statement on the low blast percentage (1-5%) and AML risk is not well-supported so should be removed.
  6. All the clinicopathologic variables discussed between lines 99 and 117 have limited prognostic value when factored against the R-IPSS parameters and the molecular mutations. Their limitation has to be stated.
  7. Lines 121-122 “As the percentage of marrow blasts roughly reflects the size of the (sub)clone that is most severely compromised in terms of cell differentiation, it is appropriately used for staging and prognostication.” This statement is not accurate biologically and clinically. There may be a very small subclone with low VAF, which can have a higher potential for leukemic transformation.
  8. The authors should refrain from using “too” at the end of their sentences (such as line 127). This is very informal English.
  9. Gene names (e.g. SF3B1) should be italicized.
  10. Lines 130-132. “As it incorporates the disease-related factors with the highest prognostic impact, the WHO classification can be used as an approximative prognostic tool”. This sentence does not add anything.
  11. Lines 133-138: How would you assess prognosis dynamically when IPSS has never been validated prospectively through the disease course of a patient?
  12. Lines 139-148: Age and gender effects are very controversial. I recommend removing these statements.
  13. Line 149: How does antecedent CHIP effect MDS prognosis? All MDS arise from CHIP!
  14. Line 150-156. The authors need to explain therapy-related MDS further including its adverse risk cytogenetic/molecular characteristics (references: 34298596, 30670442).
  15. Lines 210-223: Authors should discuss good- and poor-risk genes of IPSS-M with a special focus on TP53.
  16. Lines 278-279: Prognostic and predictive are two separate concepts that do not have to be mutually exclusive but at the same time, development of a targeted therapy does not make a predictive variable prognostic.
  17. Line 291- TPO level is not something we can measure in a routine clinical lab and it is also not validated as a predictive tool.

Author Response

(The authors gave the same response as above.)

Round 2

Reviewer 1 Report

The authors have addressed all of my concerns and I have no further comments. 

Reviewer 2 Report

My concerns have been addressed.